:ᐧ:PLOS | ONE

# Analysis of attitudinal components towards statistics among students from different academic degrees

Carmen León-Mantero[1☯¤a‡], José Carlos Casas-Rosal[2☯¤b], Alexander Maz-Machado[3¤a‡*], Miguel E. Villarraga Rico[4¤c‡]

**1** Department of Mathematics, University of Córdoba, Córdoba, Spain, **2** Department of Statistics, Econometrics, Operational Research, Business Organization and Applied Economics, University of Cordoba, Córdoba, Spain, **3** Department of Mathematics, University of Córdoba, Córdoba, Spain, **4** University of Tolima, Ibagué, Colombia

☯ These authors contributed equally to this work.
¤a Current address: Avda. San Alberto Magno, Córdoba, Spain
¤b Current address: Carretera Nacional IV, Córdoba, Spain
¤c Current address: Barrio Santa Helena Parte Alta I, Ibagué, Tolima, Colombia
‡ These authors also contributed equally to this work.
* ma1mamaa@uco.es

**Data Availability Statement:** All relevant data are within the paper and its Supporting Information files.

## Abstract

Despite its important position in academic and scientific fields, as well as in daily life, statistics is a subject that generates negative attitudes within most t disciplines in the college curriculum. This paper proposes a method for analysing different students' attitudes toward statistics using paired ANOVA tests for comparing components and groups, and discriminant analysis application for measuring the discriminant power of different components. This method was applied to a sample of 145 teachers in training from the University XXX who were studying for degrees in Spanish, English, social sciences, and mathematics during the 2016–2017 academic year. Pedagogic and anthropologic components were established using Estrada's Scale of Attitudes toward Statistics (EAEE). All the students were characterized on such a scale. The results show higher scores, mainly in instrumental components (and, to a lesser extent, cognitive and social components) from students majoring in mathematics. Furthermore, the cognitive component that most strongly characterizes students working toward a degree in social sciences, which suggests that they perceive statistics as a reliable subject but are not as aware of its utility when facing problems in everyday life. The information obtained in this study can be used to devise strategies that can lead to an improvement in future teachers' attitudes toward statistics, which would, in turn, improve the performance of their future students.

## Introduction

A knowledge of statistics provides people with the ability to make informed decisions after gathering and analyzing objective data to discriminate the veracity and falsehood of the great

**Funding:** The author(s) received no specific funding for this work.

**Competing interests:** The authors have declared that no competing interests exist.

amount of information that they receive through different mass media; to choose the principles and ideas they will adhere to regarding political, social, and cultural matters; to objectively analyze and interpret spoken and written statements; and to communicate effectively whatever information they wish to convey and proclaim [1,2]. Ridgway, Nicholson and McCusker [3] envision statistical literacy as an essential and necessary skill for people who wish to be fully functional. These skills should not considered exclusive to any one area of knowledge or profession but useful to all citizens, regardless of their education or professional profile, in understanding, tackling, and solving daily life problems.

It is especially important that teachers in all subjects and areas of knowledge acquire an adequate statistical literacy during their training in order to achieve excellence in their teaching practice, find effective ways of working with the great amount of real and objective data we all have, and provide their students with arguments and reasoning based on evidence in whatever their school environment may be. One teaching practice that holds great utility for teaching and understanding statistics involves investigative exercises. Teachers in training must know how to gather and analyze data using tables and charts, a practice that contributes to clearer presentation of arguments and complex issues, reasoning, explaining, sustaining logical arguments, and comparing and contrasting hypotheses [4].

Given the importance of statistical knowledge for prospective teachers, it is vital to pay close attention to all aspects involved in the teaching and learning of this subject—not only how teachers achieve competence in this field, but the affective aspects, such as past experiences that may have an influence on the way they teach the subject or their beliefs about statistics education [5] and, especially, the attitudes toward the subject that would affect their professional development, subject learning processes, and the attitudes of their future students [6, 7]. Although statistics is a subject of great relevance during the academic training of any student regardless of the degree taken, several studies show that students from different academic disciplines reveal different attitudes regarding the utility they consider statistics has for their academic or professional future [8–10]. Thus, this research is based on the hypothesis that the attitude an individual has towards a subject (in this case statistics) is closely linked to the academic degree that is being taken; somehow, this choice of focus reflects the preference that the student has for some subjects over others. An in-depth analysis of these factors could help teachers learn about their attitude toward their chosen degree in order propose educational innovations that could improve their attitude toward statistics based on the personalization of the teaching of this subject.

The main objective of this work is to provide a methodology for comparing the components of the attitude toward statistics teachers have depending on their major. This is applied to a sample of the prospective teachers in Spanish, English, social sciences and mathematics at the University XXX, all of whom are training to teach at secondary schools. In Colombia there are plans for training secondary school teachers as a specific university degree. There are no previous studies regarding their attitudes, but we can find such studies applied to other degrees [11], which is why we consider it necessary and relevant to conduct research emphasizing them. It is also important to highlight that the students majoring in these degrees are future middle and secondary school teachers who will be able to pursue a management position in educational institutions.

The rest of the article is structured as follows: the theoretical framework describing the attitude toward the subjects (in particular, toward statistics and its components) is presented in the next section. Then, the data collection instrument is presented, as well as a specification of the methodology. The statistical techniques used to achieve the purpose of this work are introduced, together with an overview of the sample used in this study. Then we explain the results

we obtained. Finally, we compare the results with those obtained in previous work, showing the main conclusions of our study.

## Theoretical framework

Ever since the development of the concept of attitude in education, a large and growing body of literature has investigated questions and issues from the fields of the behavioral sciences and education, although there is no consensus about theory [12]. There is, however, some agreement on certain aspects like the understanding of attitude as subjects' predisposition to behave in a particular way in specific situations, which leads to the concept of attitudes as modifiable mental states. It follows that it is possible to intervene and produce changes in people's behavior.

Different definitions of attitude can be found in the literature depending on the author's field of work and the context. Allport's [13] definition suggests that attitude is a mental state achieved through life experience and that it influences behavior. Rokeach [14] describes attitude as a series of beliefs that predispose subjects to behave in a particular way in the face of an object or situation. Aiken [15] further develops the above studies, pointing out that a behavioral response can be positive or negative. Regardless of the branch of knowledge, Hart [16] and Gómez-Chacón [17] are in agreement in understanding attitude as "an evaluative (that is, positive or negative) predisposition that determines the personal intentions and affects behavior" (p. 23). However studied and cultivated by a plethora of researchers the concept has been, it is still regarded as an ambiguous construct that requires further theoretical development. Among them are the great differences and inconsistencies found between the attitudes adopted by a subject and those that he/she claims to have [18].

Attitudes toward any particular area of knowledge (in this case, statistics), show specific characteristics: one may express enthusiasm for a subdivision of a subject and annoyance for another; even though attitudes tend to be positive at an early age, these enthusiasms may emerge at any stages or level. Attitudes evolve gradually over time, are gradable according to their intensity, and can be positive or negative. In some cases, attitudes toward the subject mirror feelings toward teachers/instructors or type of activity [19–20]. Generally, educational interventions aimed at the improvement of students' attitudes toward specific subjects have not been fruitful. All the same, there is evidence of success when bringing collaborative approaches into the classroom or employing systematic desensitization in order to reduce anxiety at an individual level [18].

The varied conceptions of the notion of attitude are composed of three elements: cognitive component, to the beliefs and conceptions about the subject of study; affective component, the feelings it arouses; and behavioral component, the behavior or inclination to react to a particular stimulus [17, 19]. In the case of attitude toward statistics, Estrada, Batanero and Fortuny [21] consider three other components: social component, its sociocultural value for any citizen; educational component, the utility and curricular difficulty of statistics; and instrumental component, the utility of statistics as compared with other areas of knowledge.

Likert scales are the instruments researchers most frequently used to collect data related to attitudes toward statistics. The following four are the most commonly used [7, 22]: Statistics Attitude Survey [SAS], Attitudes Towards Statistics Scale [ATS], and Survey of Attitudes Toward Statistics [SATS-28] and [SATS-36]. Researchers in Latin America tend to apply the following scales: Scale of Attitudes Toward Statistics [EAE] [19] and Estrada's Scale of Attitudes Towards Statistics [EAEE] [23]. The latter was designed specifically for educators and trainee teachers, combining the scales of SAS, ATS, and EAE, and including a total of 25 items. This scale was chosen for the present study.

Much of the research on the topic has focused on the analysis of attitudes toward statistics through scales observing variables such as level and type of education of the subject, age, gender, or previous statistical knowledge [20, 24–26]. The growing concern with analysing attitudes toward statistics among the different members of the educational community has produced international studies about the attitudes toward statistics in university students from the scientific, technical, and social fields [8, 11, 27], high school students [28, 29], in-service teachers [30, 31], or trainee/prospective teachers [32, 33]. Research regarding prospective teachers' attitudes toward statistics has tended to focus on students working toward degrees in primary and early years education [20, 23], pedagogy [34], and sports and physical activities science [35]. However, too little attention has been paid to prospective teachers at other levels or in other fields.

## Materials and methods

### Data collection instruments

As mentioned above, data for this study were collected using the EAEE scale of attitudes designed by Estrada [23], which has been widely used by many researchers to measure attitudes toward statistics [30, 36, 37] It is a scale with five levels of opinion, composed of 25 items (Table 1), to evaluate the following elements: pedagogic (affective, cognitive, and behavioral) and anthropologic (social, educational, and instrumental).

The scale is a Likert type, with 14 affirmative statements and 11 negative statements that allow the following scoring options: Strongly disagree = 1; Disagree = 2; Indifferent = 3; Agree = 4; Strongly agree = 5. Other than asking students to evaluate the statements in the scale, subjects were asked their gender, age, and whether and when they had received instruction in statistics as part of their formal education. Once the information was processed, those items with a negative statement were inverted to make high scores show positive attitudes.

### Statistical analysis of data

Firstly, the variables that measure the pedagogical components (affective, behavioral, and cognitive) and the anthropological components (social, educational, and instrumental) were generated in an additive way from the answers to the items that define them. Subsequently, they were standardized to values between 0 and 1 to obtain comparable values. Usually, in order to obtain the reliability of the scale, the Cronbach's alpha is computed. However, Cronbach's alpha is not a good measure of reliability; this measure has also been shown to be unrelated to a scale's internal consistency [38]. An in-depth analysis of the weaknesses of Cronbach's alpha coefficient is performed in Peters [39]. Peters also proposes the use of the omega coefficient as a substitute. This coefficient is based upon the sum of the squared loadings on the factor of the test and it measure the proportion of test variance due to the analyzed factor $\dot\omega$ in general. It can be calculated from a factor analysis output [40]. For this reason, hereafter, the reliability of

**Table 1. Components of the attitudes measured with the scale [23].**

| Pedagogic component | Anthropologic component | | |
|---|---|---|---|
| | Social | Educational | Instrumental |
| Affective | 1, 11, 25 | 7, 12, 23 | 10, 13, 16, 20 |
| Cognitive | 2, 19, 21 | 4, 6, 17 | 3, 24 |
| Behavioral | 9, 18 | 8, 15, 22 | 5, 14 |

responses to our questionnaire was analyzed by calculating the omega coefficient of the instrument and of each of the calculated components. The scale structure function of the userfriendlyscience package of R software was used [39].

Once the variables were constructed, multiple comparison contrasts were made based on the student's academic major. This was made using ANOVA contrasts after normality verification by means of the Shapiro-Wilks test and the Levine's test for homoscedasticity. This was also done by the Kruskal-Wallis test by ranks (non-parametric method), revealing the same results. Afterwards, the Duncan test for homogeneous subsets allowed us to analyze the sense of the differences observed in the different variables. In addition, the existence of differences in the various components of students studying for the same degree was analyzed by means of the analysis of variance with repeated measures, using the Greenhouser-Geisser correction, which is the most conservative. From the above students studying for a degree in mathematics demonstrated superior attitude ratings, as will be seen in the results—the elevated values showed in all of the attitudinal components led us to center the subsequent study on them.

Then, four discriminant analyses were performed using the discriminant analysis stepwise method based on Wilks' lambda. The first analysis was performed between students of mathematics and students of the rest of the degrees, while the rest of the three analyses were two-by-two analyses in order to determine the components that discriminate, in an individual way, the students majoring in Spanish, English, and the social sciences from those majoring in mathematics.

All of this allowed us to gain more in-depth knowledge of the characteristics of the attitude toward statistics that, in a more significant way, distinguish among students pursuing different degrees. To sum up: we performed comparison contrasts of components between degrees, comparison between the components of each degree and, finally, identification of the components with a greater discriminating power in each degree.

## Participants

This methodology was applied to a population of students taking initial teacher training at the formal education institutions that belong to the Faculty of Educational Sciences of the University XXX during the 2016–2017 academic year. The sample included 145 students, 29 majoring in English, 28 in Spanish, 44 in social sciences, and the rest (44 students) in mathematics. This selection was chosen to include participants studying statistics with different degrees of complexity and using that subject in diverse contexts, as well as for different purposes. In the fields of English and Spanish, the use of statistics is virtually nonexistent and the subject is not available within the curriculum; in the social sciences and mathematics statistics is a cross-curricular subject that is used to interpret results, though clearly, in the latter field the level of abstraction is superior. In this respect, and due to the proximity of the mathematics and social sciences studies with regard to this subject, one would expect to find a more positive attitude among students in these fields when compared to the other degrees, especially when it comes to analyzing the instrumental and educational components.

More than half of the sample is composed of women (55.9%), while this tendency is inverted within the degree in social sciences, where women represent up to 45% of the total. The degree in Spanish is shows a larger variance between genders, as only one out of four students were men. Equal gender distribution was observed within the degree in mathematics. The age range is wide, ranging from 16 to 34 years—the average age is 20.88 years (s = 3,394). It is worth noting that virtually all of the students who participated in this study claimed that they did not recall the material they had studied in statistics classes during their primary and middle studies—consequently, an element that could modify their answers was dismissed.

**Table 2. Omega coefficient for internal consistency analysis.**

| Attitude | Affective | Cognitive | Behavioral | Social | Educative | Instrumental |
|---|---|---|---|---|---|---|
| 0,9384 | 0,8938 | 0,8165 | 0,7865 | 0,8171 | 0,8799 | 0,8618 |

Being able to recall components such as teaching methodology used in class or their relationship with the teacher might have significantly influenced their answers.

In the study, the management of the university centre was informed about the meaning of the investigation and relevant permission was requested, which was obtained after the assessment, by means of the informed consent model that was made available by the researchers.

Data collection was carried out on four different days (one for each degree) during March 2017. Firstly, students were given a document for informed consent, according to the guidelines of the World Health Organization (attached document). This document was read aloud and all doubts and comments raised by the students were resolved. Then they were given the questionnaire and asked to assess their desire to complete it, for which they had all the time they needed; they were reminded that non-participation would not have any negative effect.

The questionnaire was completely anonymous since it did not request personal data. Each answer is coded with an alphanumeric code that identifies the degree and number of the questionnaire. The University XXX is a public center, which had approximately 20,000 students in the 2016/17 academic year. The population belonged to the middle and lower middle socioeconomic classes. The average age of the students was 20 years.

## Results

Instruments' and components' high internal consistency were confirmed by the value of the omega coefficient, which can be observed in Table 2. In every case the value is higher than 0.7, which confirms the reliability of the instrument.

The subsequent descriptive analysis shows the average scores and the standard deviations obtained for each of the components found in the answers to the questions the students from the different academic fields gave (Table 3). A comparative graphic is shown, in addition, in Fig 1.

The average values for students majoring in mathematics are higher than those of the rest of the students, with similar scores in all of the attitudinal components. These students point out that they like to teach and study statistics—they consider it useful both in the academic and professional fields and they are also aware of the importance of the subject culturally and socially.

**Table 3. Descriptive measures of the different component depending on the degree.**

| Degree | | Pedagogical Component | | | Anthropological Component | | |
|---|---|---|---|---|---|---|---|
| | | Affective | Cognitive | Behavioral | Social | Educative | Instrumental |
| Social Sciences | Mean | 0,5483 | 0,6108 | 0,5032 | 0,6484 | 0,5442 | 0,4759 |
| | Stand. Dev. | 0,1578 | 0,1642 | 0,1373 | 0,1497 | 0,1479 | 0,1529 |
| Mathematics | Mean | 0,8210 | 0,8125 | 0,7662 | 0,8054 | 0,7961 | 0,8082 |
| | Stand. Dev. | 0,1433 | 0,1345 | 0,1685 | 0,1534 | 0,1552 | 0,1366 |
| Spanish | Mean | 0,5661 | 0,5804 | 0,5026 | 0,6261 | 0,5645 | 0,4665 |
| | Stand. Dev. | 0,1771 | 0,1479 | 0,1471 | 0,1542 | 0,1925 | 0,1564 |
| English | Mean | 0,5784 | 0,6767 | 0,5172 | 0,7004 | 0,5718 | 0,4871 |
| | Stand. Dev. | 0,1952 | 0,1228 | 0,1471 | 0,1684 | 0,1806 | 0,1477 |

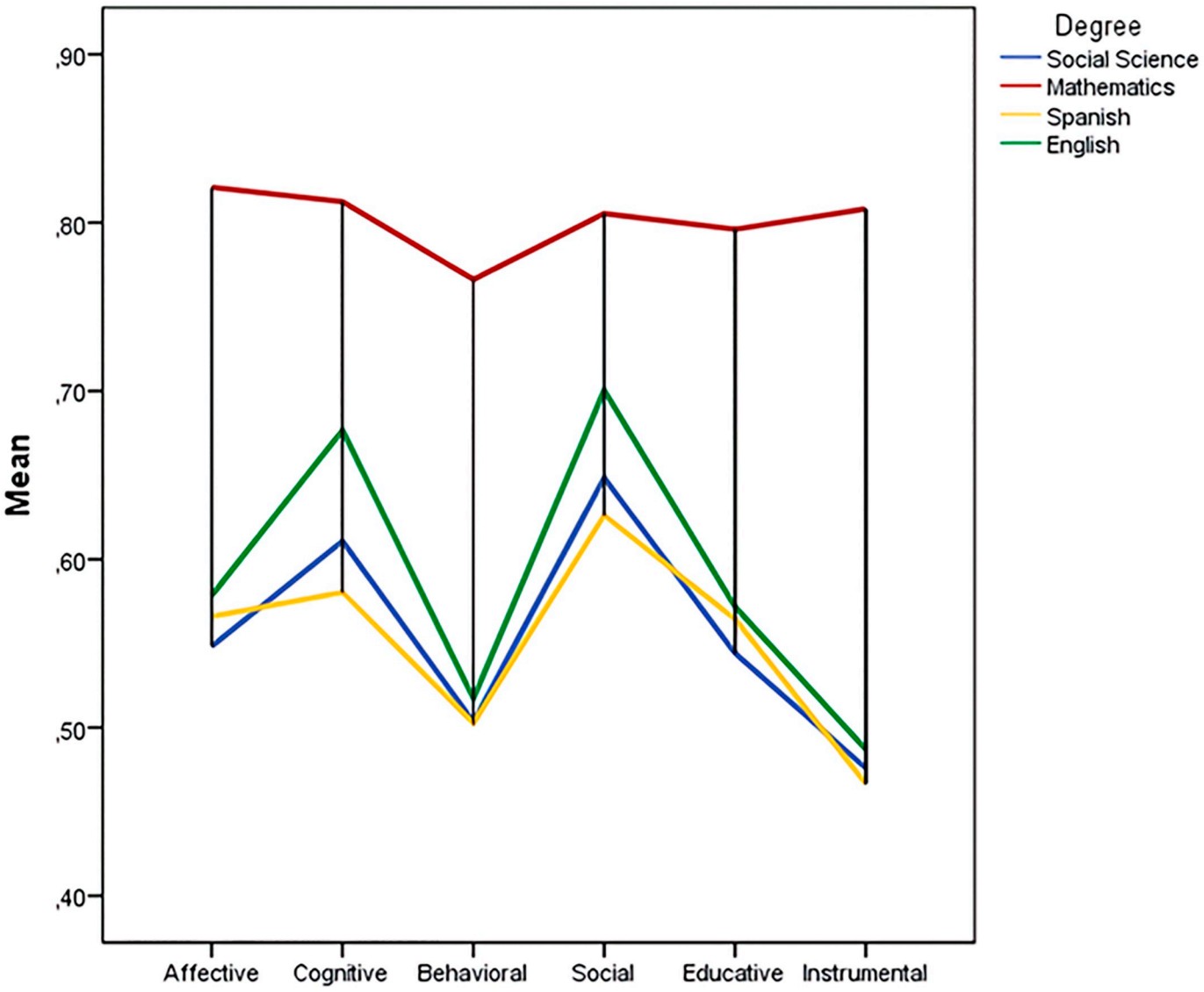

**Fig 1. Average values per degree.**

Concerning the rest of the participants, even though we can see an improvement in the social sciences component, low scores in the instrumental and behavioral component stand out. The affective and educational components remain neutral. It is noteworthy that the valuations of prospective teachers of English, Spanish, and social sciences exceed in all components, taking into account the numerous applications that can be made of statistics in the social sciences, regarding geographical, educational, economic, and political issues. Indeed, these students, among all the participants in this research, are the ones who feel least comfortable with the subject. These results suggest that, although these students understand that statistics is a useful tool that every citizen may need to face everyday problems, and that it must be studied in primary and secondary education, they value negatively the benefits that this can offer them in their academic studies or in their future professional performance. Regarding the liking for

**Table 4. Equality of variances, normality and comparison contrasts.**

| | | Pedagogical Component | | | Anthropological Component | | |
|---|---|---|---|---|---|---|---|
| | | Affective | Cognitive | Behavioral | Social | Educative | Instrumental |
| **Comparison Tests** | | | | | | | |
| **ANOVA** | Statistic | 25,219 | 20,101 | 30,578 | 10,446 | 21,224 | 51,601 |
| | p-value | 0,000 | 0,000 | 0,000 | 0,000 | 0,000 | 0,000 |
| **Kruskal Wallis** | Statistic | 54,528 | 46,914 | 52,334 | 32,940 | 45,814 | 72,417 |
| | p-value | 0,000 | 0,000 | 0,000 | 0,000 | 0,000 | 0,000 |
| **Shapiro—Wilks test for normality** | | | | | | | |
| **Social Sciences** | Statistic | 0,977 | 0,970 | 0,968 | 0,982 | 0,981 | 0,986 |
| | p-value | 0,518 | 0,298 | 0,251 | 0,697 | 0,664 | 0,858 |
| **Mathematics** | Statistic | 0,834 | 0,883 | 0,931 | 0,851 | 0,927 | 0,845 |
| | p-value | 0,000 | 0,000 | 0,011 | 0,000 | 0,008 | 0,000 |
| **Spanish** | Statistic | 0,914 | 0,939 | 0,957 | 0,827 | 0,964 | 0,967 |
| | p-value | 0,025 | 0,106 | 0,292 | 0,000 | 0,431 | 0,492 |
| **English** | Statistic | 0,968 | 0,937 | 0,966 | 0,952 | 0,967 | 0,954 |
| | p-value | 0,507 | 0,086 | 0,449 | 0,205 | 0,477 | 0,234 |
| **Equality of variance test** | | | | | | | |
| **Levene** | Statistic | 1,617 | 1,735 | 1,065 | 0,577 | 0,613 | 1,141 |
| | p-value | 0,188 | 0,163 | 0,366 | 0,631 | 0,608 | 0,335 |
| **Repeated measures ANOVA (Greenhouse-Geisser)** | | | | | | | |
| | | Social Sciences | | Mathematics | | Spanish | English |
| **Estadístico** | | 40,000 | | 4,652 | | 10,243 | 24,225 |
| **p-value** | | 0,000 | | 0,003 | | 0,000 | 0,000 |

the subject, they are not enthusiastic about the idea of working with it, but they would not eliminate it from their curriculum.

From the differences observed in each of the attitudinal components for each degree (also within the same degree), the need to contrast the existence of significant differences at the population level can be inferred through ANOVA tests and repeated measures ANOVA (parametric tests) as well as through rank-based testing (non-parametric test). Results are shown in Table 4, in which we also show the results of the Shapiro-Wilks test for normality and the Levene test for equality of variance. From the normality tests the assumption of the null hypothesis can be deduced when testing for normality in all of the contrasts except for the ones concerning the degree in mathematics, which, added to the assumption of equality of variances confirmed by the Levine test, justifies the application of ANOVA contrasts. However, the non-parametric replacement test has also been applied because of the non-realization in the degree of mathematics. Nevertheless, both tests reveal the existence of significant differences in the six attitudinal components depending on the degree that the student is taking, with p-values below 0.0001 in all cases, as well as the existence of differences between the components of every degree separately (p-value <0.004).

Two-by-two comparisons were made through the Duncan test for homogeneous subsets in order to investigate the differences more deeply. Results are shown in Table 5.

As shown at a descriptive level, the scores from the students majoring in mathematics are significantly higher in the rest of the attitudinal components. Among the other three degrees, the differences are not significant except for the cognitive component, in which the scores of the future English teachers are significantly higher than the future Spanish teachers, which suggests that the perception toward the subject is more positive among the former students. The

**Table 5. Two-by-two comparisons of both scores for different degrees.**

| Affective | Group 1 | Social Sciences | Spanish | English | |
| --- | --- | --- | --- | --- | --- |
| | Group 2 | | | | Mathematics |
| **Cognitive** | Group 1 | Spanish | Social Sciences | | |
| | Group 2 | | Social Sciences | English | |
| | Group 3 | | | | Mathematics |
| **Behavioral** | Group 1 | Spanish | Social Sciences | English | |
| | Group 2 | | | | Mathematics |
| **Social** | Group 1 | Spanish | Social Sciences | English | |
| | Group 2 | | | | Mathematics |
| **Educative** | Group 1 | Social Sciences | Spanish | English | |
| | Group 2 | | | | Mathematics |
| **Instrumental** | Group 1 | Spanish | Social Sciences | English | |
| | Group 2 | | | | Mathematics |

fact that the scores from the cognitive component of the future social science teachers are not higher than those from the degree in English stands out, as statistics may be importance in the teaching and learning of the social sciences.

The comparison of the different components of the students majoring in the same degree was also carried out by comparing means. The differences observed and the order obtained (setting a 5% significance level), are shown in Table 6. Results for social and instrumental components are similar for the degrees in social sciences, Spanish, and English—those are the components showing more discrepancies. The scores from the students majoring in mathematics do not show differences for the different components.

Finally, the components that have a higher discriminating capacity when identifying mathematics students compared to the rest was analyzed, as well as the students of each of the other degrees compared to mathematics. The aim was to determine the scores that best characterize the students of each subject.

The statistical assumptions established by Meyers, Gamst and Guarino [41] needed for the application of the discriminant analysis were checked. All the analyses were performed from a single discriminant function with a dichotomous dependent variable, and the components were included within the model by means of the stepwise method based on Wilks Lambda.

Low values of Wilks' lambda distribution and the value of chi square, with limit probabilities under 0.001 in all cases, report the low overlap of the groups that, together with the high value of the canonical correlation in all cases, allow us to draw conclusions about the discriminant capacity of the components included in each of the analyses. This is confirmed by the results of the classification, which show percentages of success between 82.10% and 97.70% (Table 7). The attitudinal component with a higher discriminant power for the students majoring in Mathematics is the instrumental component, showing that future mathematics teachers are aware of the utility statistics has for other areas of expertise: in order to provide credibility

**Table 6. Two-by-two comparison of the scores for the different degrees.**

| Degree | Components |
| --- | --- |
| **Social Sciences** | Social > Cognitive > Affective = Educative > Behav. = Instrumental |
| **Mathematics** | Affective = Cognitive = Instrumental = Social = Educative > Behav. Affective > Behavioral |
| **Spanish** | Social > Cognitive = Affective = Educative > Behav. = Instrumental |
| **English** | Social = Cognitive > Affective = Educative > Behav. = Instrumental |

**Table 7. Discriminant analysis results.**

| | | Mathematics | Spanish | English | Social Sciences |
|---|---|---|---|---|---|
| **Included Components** | | Instrumental | Instrumental | Social Instrumental | Cognitive Instrumental |
| **Autovalue** | | 1,094 | 1,367 | 1,622 | 1,488 |
| **Canonical Correlation** | | 0,723 | 0,760 | 0,787 | 0,773 |
| **Wilks' Lambda** | | 0,478 | 0,423 | 0,381 | 0,402 |
| **Chi-squared** | | 105,303 $p < 0,001$ | 59,875 $p < 0,001$ | 67,481 $p < 0,001$ | 77,472 $p < 0,001$ |
| **Standardized coefficients** | | 1,000 | 1,000 | -0,625 1,308 | -0,503 1,347 |
| **Centroids** | No | -0,686 | 0,920 | 1,020 | 1,206 |
| | Yes | 1,574 | -1,445 | -1,547 | -1,206 |
| **Correctly Classified Cases** | Case belongs to a class | 93,20% | 82,10% | 86,20% | 88,60% |
| | Case does not belong to a class | 93,10% | 93,20% | 97,70% | 93,20% |
| | Total | 93,10% | 88,90% | 93,20% | 90,90% |

in research, in order to analyse critically results obtained when solving problems, in order to help in decision-making problems, and as a tool that can help us understand complex topics. In addition, they like the subject, both in the purely academic field and in their daily lives.

The opposite result was obtained for students pursuing a degree in Spanish, these students are characterized by significantly lower scores than those obtained from the students of mathematics within the instrumental component. This is consistent with the data obtained at the descriptive level, in which prospective Spanish language teachers obtained the lowest scores in most of the attitude components except for the affective and educational components. On the other hand, students majoring in English ate characterized, in addition to significantly lower scores within the instrumental component, by high scores within the social component, which shows that they positively value the subject as a useful part of the training of all citizens because statistics can help individuals understand complex data, differentiate truthful information from false information, as well as helping to set our opinions on political, cultural, or social beliefs. Lastly, we must highlight the results obtained for the majors in social sciences. Contrary to what it was expected, the instrumental component is not the one that characterizes these students in a positive way; rather, it is the cognitive one, which suggests that they perceive statistics as a subject necessary for the basic training of all citizens, as well as for their own studies and professional development, but they are not as aware as the students majoring in the other degrees of its utility when facing problems in everyday life.

## Discussion and conclusions

This paper proposes a methodology that can be used to obtain information related to students' attitudinal components toward statistics based on the degree that they are working toward, trying to identify the relevant aspects that should be addressed when working with them to improve their attitudes toward the subject and, in turn, the attitudes of their future students. One of the objectives of this work was to identify the component of the attitude that would discriminate each group of teachers depending on the degree they were studying for. Once the characteristics of each group were detected, it was remarkable to note how those students being trained to be teachers within the field of social sciences were unaware of the utility of statistics in the realm of everyday problems. The scores of the future teachers of mathematics are higher than those taking the other degrees studied here, with the behavioral component having the lowest scores. The participants studying for the other degrees also demonstrated the lowest

scores in the instrumental and behavioral components. This may be because almost no participant in the study admitted remembering the statistical knowledge they worked on during their primary and secondary school years, since it is obvious that the knowledge that one possesses about a subject influences how he or she behaves when facing tasks, problem-solving, or decision-making related to that subject. These results are consistent with those obtained by Salinas and Mayén [29] and Estrada, Batanero and Fortuny [20] in high school students and teachers in training (primary education and preschool, respectively) using the same data-collection instrument. Both studies concluded that previous training in statistics is a decisive factor in the students' attitude toward the subject. However, in the research conducted by Ruíz de Miguel [10], attitudes towards statistics found in the students studying for degrees in pedagogy, social education, and primary and preschool education were examined using the scale designed by Auzmendi [19]. In this investigation, no differences were found regarding the attitude depending on the degree being studied for, which can be justified by the high similarity among the degrees in the study, which can lead to students' similar characteristics. The extension of this methodology to representative samples that could gather information regarding all degrees studied for would allow us to obtain valuable information that would help teachers adapt their methods of teaching statistics to customize the training according to the needs found in each subject, thus working more intensively those components that need it most.

The results obtained from the study sample highlight the need to improve the statistical knowledge of future secondary school teachers, focused on the deficiencies found in each group, which will in turn allow them to connect and work their subjects in an interdisciplinary way, as well as helping their students to carry out research exercises and, especially in the case of social science teachers, analyse data such as demographic, temperature, or climate charts. The treatment of real and contextual problems should be promoted through research questions in different disciplines.

## Author Contributions

**Conceptualization:** Carmen León-Mantero.

**Data curation:** Miguel E. Villarraga Rico.

**Formal analysis:** José Carlos Casas-Rosal.

**Investigation:** Carmen León-Mantero, José Carlos Casas-Rosal.

**Methodology:** José Carlos Casas-Rosal, Alexander Maz-Machado.

**Software:** José Carlos Casas-Rosal.

**Supervision:** Alexander Maz-Machado.

**Validation:** José Carlos Casas-Rosal.

**Writing – original draft:** Carmen León-Mantero, José Carlos Casas-Rosal.

**Writing – review & editing:** Carmen León-Mantero, Miguel E. Villarraga Rico.

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
