## [Decision Letter · Decision Letter 0]

8 Oct 2019

PONE-D-19-21019

Analysis Of Attitudinal Components Towards Statistics Among Students From Different Academic Degrees

PLOS ONE

Dear Dr. Maz-Machado,

Thank you for submitting your manuscript to PLOS ONE. After careful consideration, we feel that it has merit but does not fully meet PLOS ONE’s publication criteria as it currently stands. Therefore, we invite you to submit a revised version of the manuscript that addresses the points raised during the review process.

You need to clarify the presentation of the results. I am not so concerned about using the structural equation modelling but if it is core to your methodological framework then it is essential in the analysis and cannot be ignored in the results and discussion. You could try an approach more in line with Judea Pearl's methods.

We would appreciate receiving your revised manuscript by Nov 22 2019 11:59PM. To enhance the reproducibility of your results, we recommend that if applicable you deposit your laboratory protocols in protocols.io, where a protocol can be assigned its own identifier (DOI) such that it can be cited independently in the future. For instructions see: http://journals.plos.org/plosone/s/submission-guidelines#loc-laboratory-protocols

We look forward to receiving your revised manuscript.

Kind regards,

Andrew R. Dalby, PhD

Academic Editor

PLOS ONE

Journal Requirements:

3. In the ethics statement in the manuscript and in the online submission form, please provide additional information about participant consent. Please ensure that you have discussed whether data were collected and analyzed anonymously. Please also discuss whether participation was voluntary.

4. Please ensure that you have discussed whether all data were fully anonymized before you accessed them and/or whether the IRB or ethics committee waived the requirement for informed consent. If patients provided informed written consent to have data from their medical records used in research, please include this information.

5. We note you have included a table to which you do not refer in the text of your manuscript. Please ensure that you refer to Table 7 in your text; if accepted, production will need this reference to link the reader to the Table.

Additional Editor Comments (if provided):

Reviewers' comments:

Reviewer's Responses to Questions

**Comments to the Author**

1. Is the manuscript technically sound, and do the data support the conclusions?

Reviewer #1: Partly

2. Has the statistical analysis been performed appropriately and rigorously? 

Reviewer #1: Yes

3. Have the authors made all data underlying the findings in their manuscript fully available?

Reviewer #1: Yes

4. Is the manuscript presented in an intelligible fashion and written in standard English?

Reviewer #1: No

5. Review Comments to the Author

Reviewer #1: This paper examined the effect of the attitudes toward statistics of a sample of teachers in training in Spanish, English, Social Sciences and Mathematics. Generally, the topic is interesting and has important practical implications. However, I do have some concerns about the study, especially in the main analysis.

I strongly suggest that the manuscript need to be proofed by an English native speaker so that some grammar mistakes could be changed and the manuscript can be more readable.

Abstract: should be rewritten to mention briefly the rational as well presented in introduction, and should include more information of method (e.g. identity of participants and details of analysis) and explicitly state results (rather than mentioning the aim of the study only and leaving the readers to wonder what was found).

The structure of the introduction should be modified. The "aim" can be clarified into the introduction. And the logical connections should be strengthened among different paragraphs. Why the authors didn't use the Vancouver style for references?

Why did you not include the main structural equation modelling in the analyses? Indeed, this type of analysis it would be easier to highlight the relations between the different variables.

Results: However, the results were extremely poorly stated. Please rewrite and distill major findings to report.

6. PLOS authors have the option to publish the peer review history of their article (what does this mean?). If published, this will include your full peer review and any attached files.

Reviewer #1: No

---

## [Author Response · Author response to Decision Letter 0]

22 Nov 2019

Manuscript PONE-D-19-21019, titled “Analysis of Attitudinal Components towards Statistics among Students from different academic degrees”. 

Dear editor and reviewers, thank you very much for giving us the opportunity to revise and resubmit our article to PLOS ONE. We have found your comments to be highly helpful in improving the article in terms of the abstract, the introduction, the results and the conclusions. After carefully reading your comments, we have introduced some changes in the manuscript to address your concerns. We present our detailed comments below. 

RESPONSE TO ACADEMIC EDITOR:

0). You need to clarify the presentation of the results. I am not so concerned about using the structural equation modelling but if it is core to your methodological framework then it is essential in the analysis and cannot be ignored in the results and discussion. You could try an approach more in line with Judea Pearl's methods.

Thank you very much for the useful question, which has helped us realizing the need to better explain the methodology. We have detected an error in relation to calculating omega coefficient that has been resolved in methodology section. We have more clearly detailed the compute of the ω coefficient for estimating reliability from a factorial analysis framework (see first paragraph of “Statistical analysis of data” section): “Usually, in order to obtain the reliability of the scale, the Cronbach’s alpha is computed. However, Cronbach’s alpha is not a good measure of reliability; this measure has also been shown to be unrelated to a scale’s internal consistency [38]. An in-depth analysis of the weaknesses of Cronbach's alpha coefficient is performed in Peters [39]. Peters also proposes the use of the omega coefficient as a substitute. This coefficient is based upon the sum of the squared loadings on the factor of the test and it measure the proportion of test variance due to the analyzed factor ώ in general. It can be calculated from a factor analysis output [40]. For this reason, hereafter, the reliability of responses to our questionnaire was analyzed by calculating the omega coefficient of the instrument and of each of the calculated components. The scale structure function of the userfriendlyscience package of R software was used [39].”

1). Please ensure that your manuscript meets PLOS ONE's style requirements, including those for file naming. The PLOS ONE style templates can be found at http://www.journals.plos.org/plosone/s/file?id=wjVg/PLOSOne_formatting_sample_main_body.pdf and http://www.journals.plos.org/plosone/s/file?id=ba62/PLOSOne_formatting_sample_title_authors_affiliations.pdf

Thank you very much for the useful feedback, which has helped us realizing the need to better formatting of paper. The archives have been named according to the style requirements and the Vancouver style has been applied in citations and references. Also the format of the headings, the figure citations, the figure captions has been changed.

2). We suggest you thoroughly copyedit your manuscript for language usage, spelling, and grammar. If you do not know anyone who can help you do this, you may wish to consider employing a professional scientific editing service. 

We highly appreciate the feedback; it has led us to make substantial modifications to improve the article. The manuscript has been revised for language usage, spelling and grammar by a native English.

3). We note you have included a table to which you do not refer in the text of your manuscript. Please ensure that you refer to Table 7 in your text; if accepted, production will need this reference to link the reader to the Table.

We appreciate the constructive comment, which has been highly useful to improve the understanding of the variables involved in the study. The reference to Table 7 has been included in the new version of the article (in the “Results” section (eleventh paragraph)).

4). Please include captions for your Supporting Information files at the end of your manuscript, and update any in-text citations to match accordingly. Please see our Supporting Information guidelines for more information: http://journals.plos.org/plosone/s/supporting-information

We appreciate the constructive comment. But we do not refer to any supporting information because all the results have been included in the main manuscript tables.

RESPONSE TO REVIEWER #1

This paper examined the effect of the attitudes toward statistics of a sample of teachers in training in Spanish, English, Social Sciences and Mathematics. Generally, the topic is interesting and has important practical implications. However, I do have some concerns about the study, especially in the main analysis.

1). I strongly suggest that the manuscript need to be proofed by an English native speaker so that some grammar mistakes could be changed and the manuscript can be more readable.

We highly appreciate the feedback; it has led us to make substantial modifications to improve the article. The manuscript has been revised for language usage, spelling and grammar by a native English.

2). Abstract: should be rewritten to mention briefly the rational as well presented in introduction, and should include more information of method (e.g. identity of participants and details of analysis) and explicitly state results (rather than mentioning the aim of the study only and leaving the readers to wonder what was found).

We highly appreciate your constructive comments, which have been highly useful to improve the article in terms of its abstract. We have tried to further clarify the information with information about method and results: “Despite its important position in academic and scientific fields, as well as in daily life, statistics is a subject that generates negative attitudes within most t disciplines in the college curriculum. This paper proposes a method for analysing different students’ attitudes toward statistics using paired ANOVA tests for comparing components and groups, and discriminant analysis application for measuring the discriminant power of different components. This method was applied to a sample of 145 teachers in training from the University XXX who were studying for degrees in Spanish, English, social sciences, and mathematics during the 2016-2017 academic year. Pedagogic and anthropologic components were established using Estrada’s Scale of Attitudes toward Statistics (EAEE). All the students were characterized on such a scale. The results show higher scores, mainly in instrumental components (and, to a lesser extent, cognitive and social components) from students majoring in mathematics. Furthermore, the cognitive component that most strongly characterizes students working toward a degree in social sciences, which suggests that they perceive statistics as a reliable subject but are not as aware of its utility when facing problems in everyday life. The information obtained in this study can be used to devise strategies that can lead to an improvement in future teachers’ attitudes toward statistics, which would, in turn, improve the performance of their future students.”

3) The structure of the introduction should be modified. The "aim" can be clarified into the introduction. And the logical connections should be strengthened among different paragraphs. Why the authors didn't use the Vancouver style for references?

We highly appreciate your constructive question, which has been highly useful to improve the introduction and clarify the aim. Furthermore, the Vancouver style has been applied:

“A knowledge of statistics provides people with the ability to make informed decisions after gathering and analyzing objective data to discriminate the veracity and falsehood of the great amount of information that they receive through different mass media; to choose the principles and ideas they will adhere to regarding political, social, and cultural matters; to objectively analyze and interpret spoken and written statements; and to communicate effectively whatever information they wish to convey and proclaim [1,2]. Ridgway, Nicholson and McCusker [3] envision statistical literacy as an essential and necessary skill for people who wish to be fully functional. These skills should not considered exclusive to any one area of knowledge or profession but useful to all citizens, regardless of their education or professional profile, in understanding, tackling, and solving daily life problems . 

It is especially important that teachers in all subjects and areas of knowledge acquire an adequate statistical literacy during their training in order to achieve excellence in their teaching practice, find effective ways of working with the great amount of real and objective data we all have, and provide their students with arguments and reasoning based on evidence in whatever their school environment may be. One teaching practice that holds great utility for teaching and understanding statistics involves investigative exercises. Teachers in training must know how to gather and analyze data using tables and charts, a practice that contributes to clearer presentation of arguments and complex issues, reasoning, explaining, sustaining logical arguments, and comparing and contrasting hypotheses [4].

Given the importance of statistical knowledge for prospective teachers, it is vital to pay close attention to all aspects involved in the teaching and learning of this subject—not only how teachers achieve competence in this field, but the affective aspects, such as past experiences that may have an influence on the way they teach the subject or their beliefs about statistics education [5] and, especially, the attitudes toward the subject that would affect their professional development, subject learning processes, and the attitudes of their future students [6, 7]. Although statistics is a subject of great relevance during the academic training of any student regardless of the degree taken, several studies show that students from different academic disciplines reveal different attitudes regarding the utility they consider statistics has for their academic or professional future [8-10]. Thus, this research is based on the hypothesis that the attitude an individual has towards a subject (in this case statistics) is closely linked to the academic degree that is being taken; somehow, this choice of focus reflects the preference that the student has for some subjects over others. An in-depth analysis of these factors could help teachers learn about their attitude toward their chosen degree in order propose educational innovations that could improve their attitude toward statistics based on the personalization of the teaching of this subject.

The main objective of this work is to provide a methodology for comparing the components of the attitude toward statistics teachers have depending on their major. This is applied to a sample of the prospective teachers in Spanish, English, social sciences and mathematics at the University XXX, all of whom are training to teach at secondary schools. In Colombia there are plans for training secondary school teachers as a specific university degree. There are no previous studies regarding their attitudes, but we can find such studies applied to other degrees [11], which is why we consider it necessary and relevant to conduct research emphasizing them. It is also important to highlight that the students majoring in these degrees are future middle and secondary school teachers who will be able to pursue a management position in educational institutions.”

4) Why did you not include the main structural equation modelling in the analyses? Indeed, this type of analysis it would be easier to highlight the relations between the different variables.

Thank you very much for the useful question, which has helped us realizing the need to better explain the methodology. We have detected an error in relation to calculating omega coefficient that has been resolved in methodology section. We have more clearly detailed the compute of the ω coefficient for estimating reliability from a factorial analysis framework (see first paragraph of “Statistical analysis of data” section): “Usually, in order to obtain the reliability of the scale, the Cronbach’s alpha is computed. However, Cronbach’s alpha is not a good measure of reliability; this measure has also been shown to be unrelated to a scale’s internal consistency [38]. An in-depth analysis of the weaknesses of Cronbach's alpha coefficient is performed in Peters [39]. Peters also proposes the use of the omega coefficient as a substitute. This coefficient is based upon the sum of the squared loadings on the factor of the test and it measure the proportion of test variance due to the analyzed factor ώ in general. It can be calculated from a factor analysis output [40]. For this reason, hereafter, the reliability of responses to our questionnaire was analyzed by calculating the omega coefficient of the instrument and of each of the calculated components. The scale structure function of the userfriendlyscience package of R software was used [39].”

5). Results: However, the results were extremely poorly stated. Please rewrite and distill major findings to report.

We highly appreciate your constructive comment, which has been highly helpful to improve the results: 

“The affective and educational components remain neutral. It is noteworthy that the valuations of prospective teachers of English, Spanish, and social sciences exceed in all components, taking into account the numerous applications that can be made of statistics in the social sciences, regarding geographical, educational, economic, and political issues. Indeed, these students, among all the participants in this research, are the ones who feel least comfortable with the subject. These results suggest that, although these students understand that statistics is a useful tool that every citizen may need to face everyday problems, and that it must be studied in primary and secondary education, they value negatively the benefits that this can offer them in their academic studies or in their future professional performance. Regarding the liking for the subject, they are not enthusiastic about the idea of working with it, but they would not eliminate it from their curriculum”

…

“In addition, they like the subject, both in the purely academic field and in their daily lives.”

…

“This is consistent with the data obtained at the descriptive level, in which prospective Spanish language teachers obtained the lowest scores in most of the attitude components except for the affective and educational components”

---

## [Editor Report · Decision Letter 1]

16 Dec 2019

Analysis Of Attitudinal Components Towards Statistics Among Students From Different Academic Degrees

PONE-D-19-21019R1

Dear Dr. Maz-Machado,

We are pleased to inform you that your manuscript has been judged scientifically suitable for publication and will be formally accepted for publication once it complies with all outstanding technical requirements.

With kind regards,

Andrew R. Dalby, PhD

Academic Editor

PLOS ONE
---

## [Editor Report · Acceptance letter]

27 Dec 2019

PONE-D-19-21019R1 

Analysis Of Attitudinal Components Towards Statistics Among Students From Different Academic Degrees 

Dear Dr. Maz-Machado:

I am pleased to inform you that your manuscript has been deemed suitable for publication in PLOS ONE. Congratulations! Your manuscript is now with our production department. 

With kind regards,

on behalf of

Dr. Andrew R. Dalby 

Academic Editor

PLOS ONE